

# Phylogenetic and population genetic analyses of *Thrips tabaci* Lindeman (Thysanoptera: Thripidae) on *Allium* host in India

Tushar Gawai[1,2], Sharwari Sadawarte[3], Kiran Khandagale[1], Anusha Raj[1], Abhijeet Kulkarni[3], Durgesh Kumar Jaiswal[1,2], Avinash B. Ade[2] and Suresh Gawande[1]

[1] ICAR-Directorate of Onion and Garlic Research, Pune, India
[2] Department of Botany, Savitribai Phule Pune University, Pune, India
[3] Department of Bioinformatics, Savitribai Phule Pune University, Pune, India

Corresponding author
Suresh Gawande,
Suresh.Gawande@icar.gov.in

## ABSTRACT

**Background**. Onion thrips (*Thrips tabaci*) is a complex of cryptic species with subtle morphological differences and distinct genetic backgrounds; thus, species identification using traditional methods remains challenging. The existence of different haplotypes and genotypes within a species can significantly influence various aspects of its biology, including host preference, reproductive capacity, resistance to pesticides, and vector competence for plant viruses. Understanding the genetic diversity and population structure of cryptic species within *T. tabaci* will not only aid in the development of more effective control strategies tailored to specific genetic variants but also in monitoring population dynamics, tracking invasive species, and implementing quarantine measures to prevent the spread of economically damaging thrips biotypes.

**Methods**. This study aims to explore intraspecies genetic diversity and molecular evolutionary relationships of the mitochondrial cytochrome oxidase gene subunit I (mtCOI) in *T. tabaci* populations from India. To capture diversity within the Indian *T. tabaci* populations, amplicon sequencing was performed for the thrips mtCOI gene from eight diverse localities in India. A total of 48 sequences retrieved for the mtCOI gene from the NCBI Nucleotide database were analysed.

**Results**. Multiple insertions and deletions were detected at various genomic positions across the populations from different localities, with the highest variation observed in the 300–400 genome position range. Molecular diversity analyses identified 30 haplotypes within the population, with certain subpopulations exhibiting higher gene flow. Analysis of single nucleotide polymorphism patterns within the mtCOI gene across diverse Indian locales revealed significant intrapopulation genetic heterogeneity and its potential repercussions on gene functionality. Elevated F statistics ($F_{st}$) values in the northern–western subpopulations suggested high genetic variability, particularly evident in haplotype networks originating mainly from the northern region, notably Delhi. While most populations displayed stable and ancient evolutionary histories, thrips populations from northern, western, and north-eastern regions indicated rapid growth.

Phylogenetic analysis

## INTRODUCTION

Thrips is a widespread polyphagous insect pest from the order Thysanoptera. Globally, several species of thrips cause serious damage to crops (*Mouden et al., 2017*; *Stuart, Gao & Lei, 2011*; *Raut et al., 2020*). Among these, the onion thrips, *Thrips tabaci* Lind, is an important pest that costs significant economic losses to various crops every year (*De Kogel et al., 2015*). Although many thrips have been reported in Alliums, *T. tabaci* is most commonly observed (*Porta et al., 2023*). In addition to direct damage to the crop, this insect serves as a vector of two devastating viruses, Iris Yellow Spot Virus (IYSV) and Tomato Spotted Wilt Virus (TSWV) (*Komondy et al., 2024*; *Nachappa et al., 2020*). Moreover, damage due to feeding by *T. tabaci* increases susceptibility to infections by other pathogens in onion (*Leach et al., 2019*; *Grode et al., 2017*). *T. tabaci* is a complex of genetically differentiated and widespread insect pests that are adapted to diverse climatic conditions (*Brunner et al., 2004*; *Gill et al., 2015*). The genetic variation within this species positively influences their adaptability under diverse climatic conditions. This genetic variation is also linked to their reproductive modes, such as thelytoky, arrhenotoky, and deuterotoky (*Kobayashi & Hasegawa, 2012*; *Nault et al., 2006*; *Gawande et al., 2017*; *Porta et al., 2023*), as well as vector competence (*Jacobson & Kennedy, 2013*).

To study the genetic variation in thrips, various genes have been targeted and analysed for phylogenetic analyses and species identification (*Buckman, Mound & Whiting, 2013*; *Rebijith et al., 2013*). DNA barcoding is a novel system of species identification; it uses the mitochondrial cytochrome C oxidase subunit 1 gene (cox1 or COI) as a standardized single molecular marker for the identification, comparison, and categorisation of animal species (*Hebert et al., 2003*). The DNA barcoding method has a prerequisite of linking species identification to a voucher from a curated biological collection. This enables follow-up and the development of a method for verifying species identification.

In Thysanopetra, targeting mitochondrial genome is the most suitable approach for molecular identification as well as species recognition irrespective of life stages, sex, and polymorphism (*Asokan et al., 2007*), identifying cryptic species (*Glover et al., 2010*; *Rebijith et al., 2013*), biotypes (*Shufran et al., 2000*), and haplotypes and host- and geographic-associated genetic differences (*Rebijith et al., 2013*; *Brunner et al., 2004*). Three distinct lineages (T-Tobacco associated, L1 and L2-Leek associated) in *T. tabaci* based on mitochondrial data were proposed (*Brunner et al., 2004*; *Kobayashi, Yoshimura & Hasegawa, 2013*; *Jacobson et al., 2016*). Using mitochondrial cytochrome oxidase I (mtCOI) sequences, microsatellite markers and vector competence were documented in accordance with the transmission efficiency of viruses in thrips population (*Jacobson & Kennedy, 2013*). DNA barcoding using mtCOI gene for species-level identification has received widespread support as an additional technique to resolve taxonomic ambiguities. According to the International Barcode of Life Consortium, genetic variation below the level of a species

has been estimated using mitochondrial genes. Its importance in quick and reliable species identification has been widely acknowledged in a numerous taxa worldwide (*International Barcode of Life Consortium, 2023*). This method has also been applied for thrips identification, diversity analysis (*Bravo-Pérez et al., 2018*), phylogenetic analysis (*Gawande et al., 2017*), population structure (*Porta et al., 2023*), and invasive genetics (*Tyagi et al., 2017*).

In India, phylogenetic analyses based on the mtCOI gene were carried out, revealing different biotypes of thrips on different hosts and exhibiting differences in their mtCOI gene (*Asokan et al., 2007*). The existence of heteroplasmy was identified in *T. tabaci* by analysing mtCOI haplotypes from different geographic regions in India (*Gawande et al., 2017*). However, an exhaustive analysis of genetic diversity and population genetic variation of *T. tabaci* in India is yet to be conducted. Its correlation with data from other countries also needs to be studied. No contemporary Indian research has reported population genetics analysis on *T. tabaci*. Despite extensive studies on thrips genotyping using mitochondrial mtCOI markers, the extent of crypticness within *T. tabaci* remains unknown. With the advent of next-generation sequencing, it has become easier to characterize the level of variation through amplicon sequencing of the mtCOI gene in a population. This technique allows for the processing and characterization of a large amount of sequence data, simplifying the capture of sequence variations within a single gene fragment. Herein, we aimed to unravel the genetic variation and structure of *T. tabaci* on *Allium* hosts from different geographic locations across India using amplicon sequencing of the mtCOI gene. We characterized the extent of genetic cryptic nature of the onion-infesting *T. tabaci* population in India using advanced bioinformatics tools.

## MATERIALS & METHODS

### Ethics statement
*Thrips tabaci* sampling from *Allium* host did not involve any endangered species.

### Specimen collection
*T. tabaci* adults were collected from *Allium* hosts at 10 locations across different climatic regions in India from 2017 to 2019, as outlined in Table 1. The sampling sites were chosen to encompass maximum agro-climatic zones where onions are widely cultivated. *T. tabaci* identification was confirmed based on taxonomic characteristics such as grey ocellar pigmentation, brown antennal segments III–IV with the basal half pale, forewings being pale, and 7-segmented antennae. Collected specimens were preserved in 95% ethanol and stored at −20 °C prior to DNA extraction.

### PCR amplification of a COI fragment and amplicon sequencing
Thrips were removed from the ethanol stock by using a fine brush. Preserved thrips were dried for 5 min and total genomic DNA was extracted using a DNeasy Blood & Tissue Kit (Qiagen GmbH, Hilden, Germany) from 50 adult thrips of each location, as per protocol described by the manufacturer. A 525-bp fragment of the mitochondrial cytochrome oxidase I was amplified using the primer pair COI-1F TGTTACGGCTCACGCTTTTG and

**Table 1** Detail information of collection of *Thrips tabaci* from India.

| Agroclimatic Zone | State | Collection site | Longitude (E) | Latitude (N) |
|---|---|---|---|---|
| I | Himachal Pradesh | Palampur | 32°06′02.0″N | 76°32′47.6″E |
| III | Sikkim | Gangtok | 27°19′11.1″N | 88°36′07.4″E |
| | Tripura | Lembucherra | 23°54′24.0″N | 91°18′47.6″E |
| VI | Delhi | IARI | 28°38′10.6″N | 77°9′27.5″E |
| | Haryana | Karnal | 29°44′55.4″N | 76°59′48.5″E |
| | Gujarat | Junagadh | 21°30′22.7″N | 70°26′57.8″E |
| VII | Maharashtra | Rajgurunagar | 18°50′34.6″N | 73°53′04.8″E |
| VIII | Tamilnadu | Coimbatore | 11°0′32.2″N | 73°56′4.33″E |

COI-3R-TAAAACAGGGTCCCCTCCCC (*Gawande et al., 2017*). PCR was performed in a 20-µL reaction volume, with the following cycling conditions: 95 °C (3 min), 35 cycles at 95 °C (40 s), 48 °C (1 min), 72 °C (2 min); and final extension at 72 °C (5 min). The PCR reaction yielded a 525-bp amplicon. PCR products from each locality were examined through 1.5% agarose gel electrophoresis. They were purified using a PCR purification kit (Qiagen); equal concentration of each sample was sent for amplicon sequencing to the commercial firm. Amplicon sequencing was performed at AgriGenome Labs Private Limited (Kerala, India) on an Illumina MiSeq 250 × 2 platform (Illumina, Inc., San Diego, CA, USA). Quality parameters such as base quality score distributions, average base content per read, and GC distribution were examined for over-represented sequences, and adapter trimming was used for quality analysis.

## Single nucleotide polymorphism analysis

Single nucleotide polymorphism (SNP) analysis started with a quality assessment of the raw amplicon sequence data by using FASTQC (https://github.com/s-andrews/FastQC/releases/tag/v0.12.1), ensuring data integrity for further analysis. Sequences were aligned using Bowtie2 (https://bowtie-bio.sourceforge.net/bowtie2/index.shtml), with a specific isolate as the reference sequence (KF724977.1), establishing a standard for variation detection. SNP calling was performed using BCFtools mpileup (https://samtools.github.io/bcftools/bcftools.html), meticulously identifying SNPs by analyzing sequence read quality and coverage depth. This analysis unveiled disparity in the variable sites between two datasets, A and B, which had 40 and 13 variable sites, respectively; a higher divergence rate was observed in dataset B despite shorter sequence lengths. This suggested that sequence length influences the perceived genetic variation, highlighting the intricate relationship between genetic sequence characteristics and evolutionary dynamics.

## Preparing a dataset for genetic diversity analyses

To elucidate genetic diversity among Indian species, an analytical study was conducted focusing on mtCOI gene sequences of Indian species retrieved from the NCBI-GenBank database. Acknowledging the inherent variability in sequence lengths—a reflection of genetic diversity—the dataset was categorised into two groups based on nucleotide length for refined computational analysis. Group A contained 25 sequences, each up to 516

| Table 2 Distribution of Group A. | | |
|---|---|---|
| **Accession ID** | **Geographical location** | **Sub-population group** |
| KF724977.1 | Maharashtra | |
| KJ868776.1 | Maharashtra | |
| KJ868777.1 | Maharashtra | |
| KJ868778.1 | Maharashtra | Sub-population 1: WEST |
| KT427421.1 | Maharashtra | |
| KJ868793.1 | Gujarat | |
| KJ868792.1 | Rajasthan | |
| KJ868791.1 | Rajasthan | |
| KJ868779.1 | Karnataka | |
| KJ868780.1 | Karnataka | Sub-population 2: SOUTH |
| KJ868781.1 | Karnataka | |
| KJ868782.1 | Tamilnadu | |
| KJ868783.1 | Odisha | |
| KJ868784.1 | West Bengal | Sub-population 3: EAST |
| KJ868786.1 | Bihar | |
| KJ868788.1 | Haryana | |
| KJ868789.1 | Punjab | |
| KJ868790.1 | Jammu Kashmir | |
| 303 | Haryana | Sub-population 4: NORTH |
| 304 | Delhi | |
| 307 | Himachal Pradesh | |
| KJ868794.1 | Madhya Pradesh | Sub-population 5: CENTRAL |
| KJ868787.1 | Uttar Pradesh | |
| KJ868785.1 | Manipur | Sub-population 6: NORTH-EAST |
| 308 | Sikkim | |

nucleotides in length, whereas Group B comprised 22 sequences, each having a length of 675 nucleotides (Tables 2 and 3, respectively). This division was strategic, catering to the computational demands posed by sequence length variability.

The MAFFT (https://mafft.cbrc.jp/alignment/server/) tool was used for multiple sequence alignment (MSA), and high-quality alignment was achieved by optimizing sequence homology and minimizing alignment gaps. Thereafter, genetic distance was calculated using the uncorrected p-distance model in MEGA-X, providing a direct measure of genetic divergence without presupposing any evolutionary model. Phylogenetic relationships among the sequences were deciphered using IQ-TREE software (*Trifinopoulos et al., 2016*), employing the maximum likelihood method with the Jukes-Cantor model (*Tamura et al., 2013*) of nucleotide substitution. This method is crucial for determining the most probable evolutionary relationships among the species. Further, genetic diversity was analyzed using DNASP v.5 (*Librado & Rozas, 2009*) and Arlequin 3.5 (*Excoffier & Lischer, 2010*) tools, aimed at evaluating genetic variation and structure within and between populations.

| Table 3 | Distribution of Group B. | |
|---|---|---|
| **NCBI GenBank Accession ID** | **Geographical location** | **Sub-population** |
| MN594551.1 | Delhi | |
| MT991561.1 | Delhi | Sub-population 1: NORTH |
| MT992026.1 | Delhi | |
| MN972625.1 | Karnataka | |
| MN972630.1 | Karnataka | |
| KF015428.1 | Karnataka | |
| KF015429.1 | Karnataka | |
| KF015430.1 | Karnataka | Sub-population 2: SOUTH |
| KF015431.1 | Karnataka | |
| KF015432.1 | Karnataka | |
| KF015433.1 | Karnataka | |
| KF015434.1 | Karnataka | |
| 306 | Tamilnadu | |
| KX622444.1 | West Bengal | |
| KX622445.1 | West Bengal | |
| KX622446.1 | West Bengal | Sub-population 3: EAST |
| KX622447.1 | West Bengal | |
| KX622443.1 | West Bengal | |
| 309 | Tripura | |
| 302 | Gujrat | Sub-population 3: WEST |
| 305 | Maharashtra | |

## RESULTS

### Sequencing data

Data generated from amplicon sequencing were used for phylogenetic analysis. The sequence produced by Illumina sequencing was used to identify the homologs of the DOGR isolates by using NCBI BLAST. The results obtained from BLAST were considered to allocate the closest homologs of DOGR new isolate. These homologs were used for the phylogenetic analysis. Statistics of a raw amplicon sequencing are presented in Table 4. The highest and lowest number of reads were generated in samples from Sikkim (247,362) and Tamil Nadu (204,698), respectively. GC content was 33.6–34.5%. More than 84% reads secured a Phred score of 30, indicating high-quality data. The Phred quality score numerically expresses the accuracy of each nucleotide; the higher the Q number, the greater is the accuracy of data. Raw sequencing data were submitted to the NCBI SRA database (PRJNA917098).

### Genomic distribution of SNPs

We obtained partial (420 bp) mtCOI sequences for *T. tabaci* sampled from *Allium* hosts from eight climatic regions of India. SNPs within the partial COI gene differentiated *T. tabaci* populations into 30 mtDNA haplotypes in total, with 17 and 13 haplotypes in groups A and B, respectively.

**Table 4** Raw amplicon sequencing data.

| Localities | Total read bases (bp) | Total reads | GC(%) | AT(%) | Q20(%) | Q30(%) |
|---|---|---|---|---|---|---|
| Junagadh | 71,282,820 | 236,820 | 34.04 | 65.96 | 92.73 | 86.52 |
| Karnal | 70,529,116 | 234,316 | 34.53 | 65.47 | 90.5 | 84.24 |
| Delhi | 66,422,874 | 220,674 | 34.05 | 65.95 | 93.34 | 87.39 |
| Maharashtra | 64,141,294 | 213,094 | 33.87 | 66.13 | 94.22 | 88.41 |
| Tamilnadu | 61,614,098 | 204,698 | 34.37 | 65.63 | 91.73 | 85.59 |
| Palampur | 66,067,694 | 219,494 | 33.61 | 66.39 | 93.62 | 87.75 |
| Sikkim | 74,455,962 | 247,362 | 33.7 | 66.3 | 95.25 | 89.83 |
| Tripura | 67,244,604 | 223,404 | 33.73 | 66.27 | 95.04 | 89.25 |

Locality-wise percent variation was calculated from the total reads recorded (Fig. 1). 307_Palampur represented the highest nucleotide polymorphism, followed by 306_Tamilnadu, 305_Maharashtra, 303_Haryana, and 302_Gujarat. Compared to other localities, a low level of mitochondrial genetic variation was observed in 304_Delhi isolates. We examined the frequency and distribution of SNPs at different locations in India. Numerous insertions and deletions of varying lengths were found in the genomic positions of different localities. For each locality, the highest and lowest variations were observed at the genome positions 300–400 and 150–250, respectively. Thus, the range of diversity at nucleotide level reported within mtCOI gene was 1–420 bp, which was used herein *via* high throughput sequencing (Fig. 2A). The top three SNPs in each 50-bp interval of mtCOI gene sequence data are shown in Fig. 2B. More detailed representation of SNP distribution in the mtCOI sequences of different samples is presented in Fig. S1.

## Molecular phylogenetic analysis

Phylogenetic trees confirmed that molecular diversity in the same geographical region is possible if the environmental conditions are favourable. Molecular evolution might not be affected by geographical proximity in various subpopulations because sequences have significantly different clusters (*i.e.,* they are not clustered as per their geographical locations). Among phylogenetic trees, most sequences formed individual nodes, suggesting highly diverse populations. As shown in Fig. 3, northern and north-eastern (Sikkim strain) subpopulations showed similarities because they formed one clade, which is isolated from the central region of India (Madhya Pradesh); similarly, the southern subpopulation (Karnataka) formed one clade, indicating the presence of a uniformly evolving population. Because of diversity in the north-eastern subpopulation, they were considered two differently grouped nodes. Similarly, strains from eastern India (Odisha and West Bengal as well as Bihar) were clustered differently, despite close geographical locations.

As shown in Fig. 4, newly sequenced isolates were closely related, irrespective of geographical locations, indicating stabilization of the thrips population (302_Gujrat, 305_Maharashtra, 309_Tripura, 306_Tamilnadu). Strains from West Bengal showed uniform compositions because they were grouped with different geographical locations,

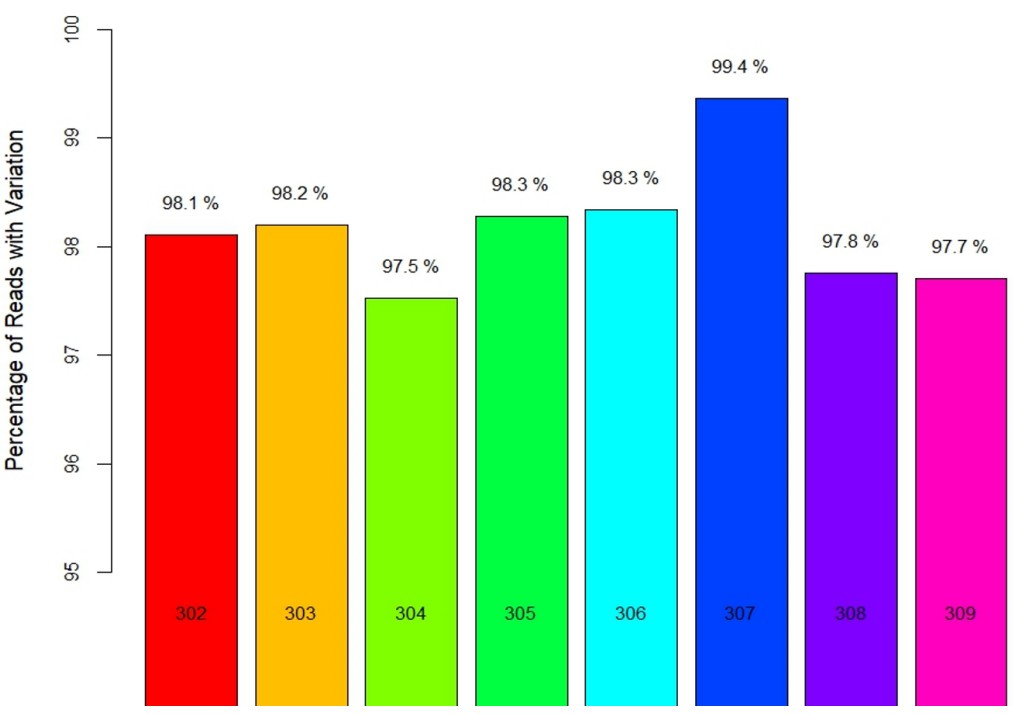

**Figure 1** **Locality wise percent reads with variations.**

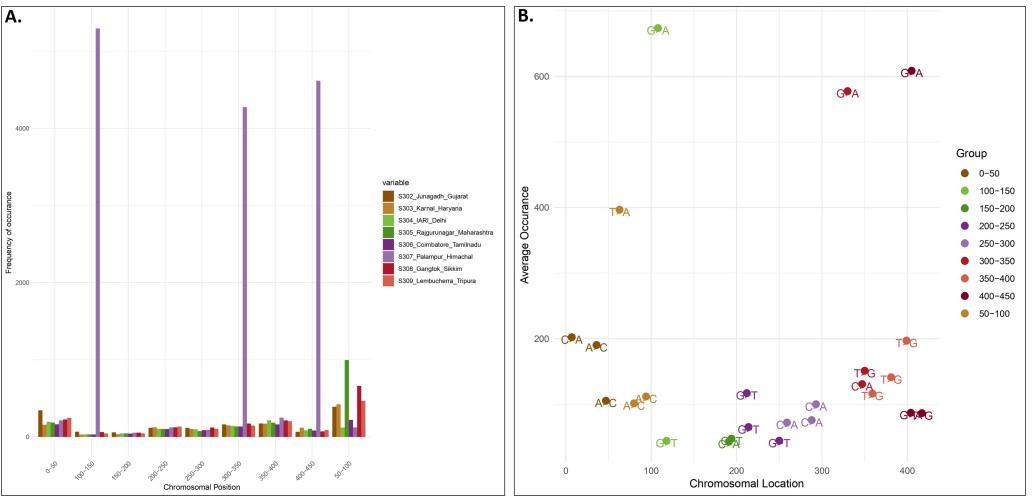

**Figure 2** **SNPs distribution *vs* Genome position.** (A). The *x*-axis corresponds to the genomic position of each SNP. The *y*-axis indicates the frequency of occurence of each SNP. Each colour corresponds to a sample collected from different localities.) (B). Distribution of top 3 SNPs at the different position of mt-COI fragment (denoted by different colour).

whereas majority of strains isolated from Karnataka showed more diverse compositions. Most populations formed individual nodes, indicating increasing molecular diversity in Group B as well.

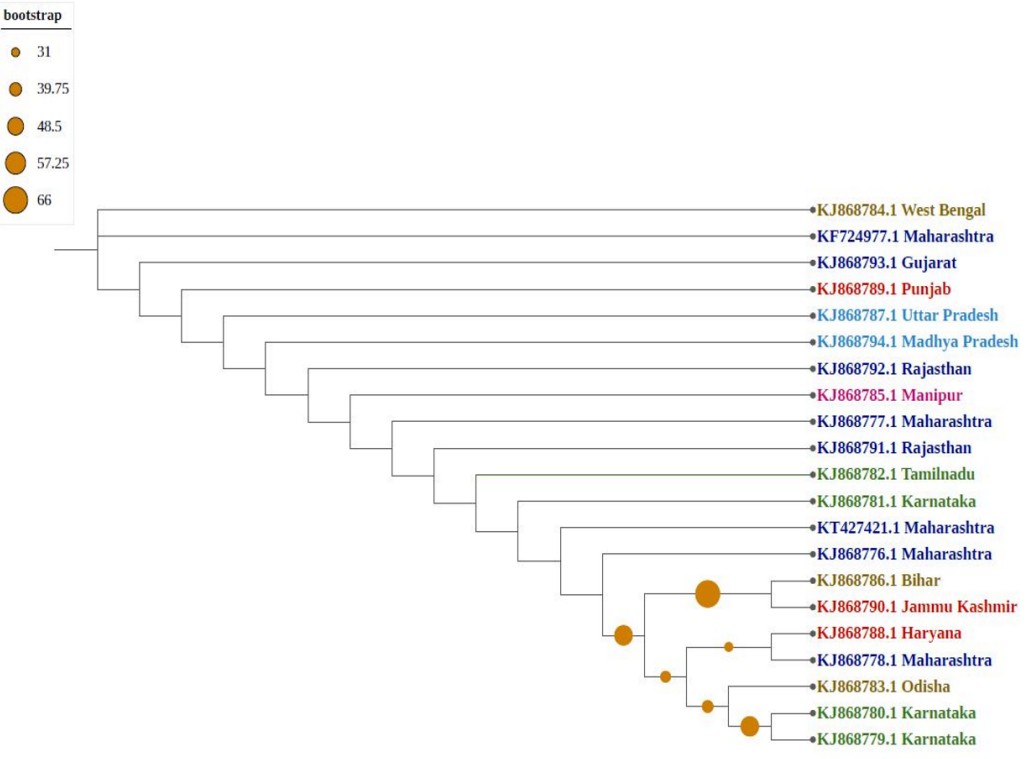

**Figure 3** **Phylogenetic analysis of Group A using Maximum likelihood method.** Violet represents sequences from northern India, blue from southern India, red from western India, green from eastern India, pink from north-eastern India, and yellow from central part of India.

## Genetic diversity analysis

The study of polymorphism in DNA sequences is a prevailing method for elucidating genetic diversity and functional relevance of these genetic variations. Genetic diversity parameters such as number of polymorphic sites (S), number of haplotypes (H), haplotype diversity (Hd), nucleotide diversity (Pi), and average number of nucleotide differences (K) were determined using DnaSP 5.12.01 (*Librado & Rozas, 2009*). Sequence analysis of Group A suggested that the population is stable with long evolutionary history and high Hd and Pi values (Hd > 0.5 and Pi > 0.005) (Table 5). Eastern populations of groups A and B as well as the northern population of Group A showed similar characteristics. Southern population of Group A had high Hd and low Pi values (Hd > 0.5 and Pi < 0.005), representing population bottleneck followed by rapid growth in population File S1). Western population of Group A showed recent population bottleneck or very few mtDNA lineages, as they had low Hd and Pi values (Hd < 0.5 and Pi < 0.005) (*Grant & Bowen, 1998*).

Higher Nm values indicating high gene flow were observed more frequently across populations in Group A than in Group B. North-eastern and eastern Indian subpopulations with respect to northern and southern subpopulations from Group A and western population with northern subpopulation from Group B had Nm values <1, indicating

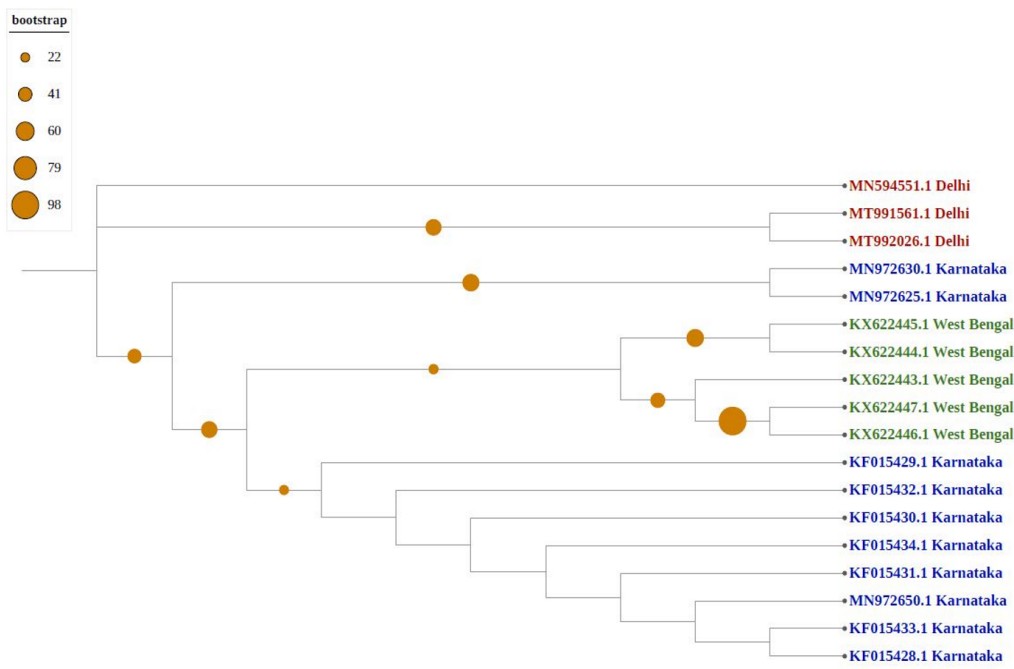

**Figure 4   Phylogenetic analysis of Group B using Maximum likelihood method.** Blue represents sequences from southern India, green from eastern India, red from Northern India, and pink from western India.

**Table 5   Genetic diversity analysis of group A and B.**

| Population | Sequences analyzed | Number of polymorphic sites (S) | Haplotype diversity (Hd) | Nucleotide diversity (Pi) | Average number of nucleotide differences (K) | Number of haplotypes | Group |
|---|---|---|---|---|---|---|---|
| All sequences | 25 | 640 | 0.69 | 0.46 | 74.94 | 17 | |
| WEST | 8 | 3 | 0.25 | 0.0014 | 0.75 | 2 | |
| SOUTH | 4 | 2 | 0.83 | 0.0022 | 1.16 | 3 | |
| EAST | 3 | 11 | 1.00 | 0.014 | 7.33 | 3 | A |
| NORTH | 6 | 640 | 1.00 | 1.65 | 195.4 | 6 | |
| CENTRAL | 2 | 0 | 0.00 | 0 | 0.00 | 1 | |
| NORTH EAST | 2 | 499 | 1.00 | 1.16 | 247.00 | 2 | |
| All sequences | 22 | 1,035 | 0.87 | 0.20 | 62.72857 | 13 | |
| NORTH | 3 | 78 | 1.00 | 0.25 | 0.33 | 3 | |
| SOUTH | 11 | 584 | 0.49 | 0.12 | 59.09 | 4 | B |
| EAST | 5 | 20 | 9.0 | 0.00683 | 4.40 | 4 | |
| WEST | 2 | 666 | 1.0 | 0.39333 | 118.00 | 2 | |

low gene flow and potentially higher genetic differences in these subpopulations within specific geographical territories. Fst values were <0.5, implying overall similarity between various subpopulations of groups A and B. The low Fst values among western, southern,
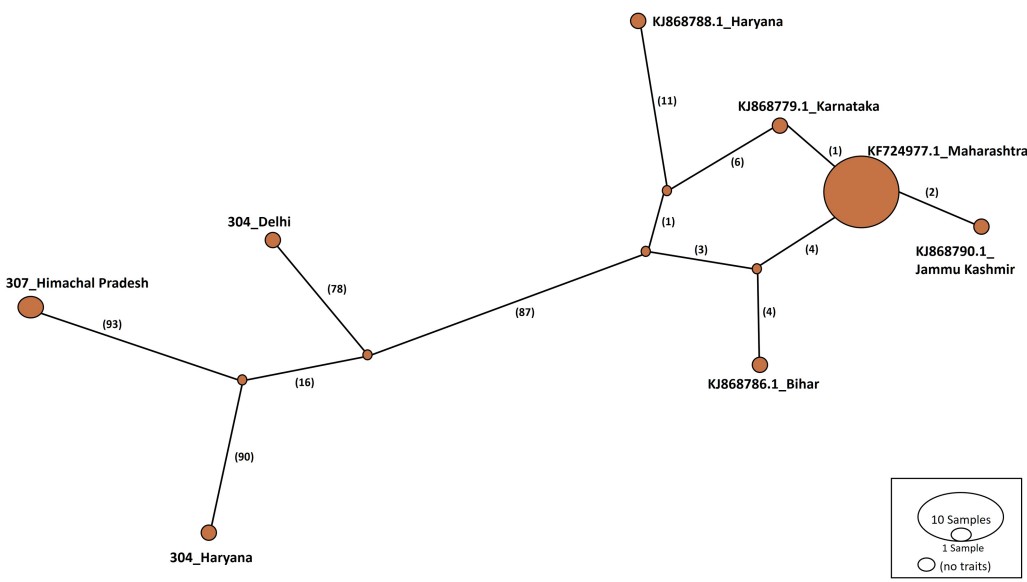

**Figure 5** **Haplotype network visualized in POPART for group A.** Numbers in bracket on the edges signify number of mutations and KF724977.1-Maharashtra represents predominant haplotype. Network has 8 haplotypes.

and northern subpopulations, along with north-eastern subpopulations in Group A, indicated a low degree of differentiation among both subpopulations.

Haplotype networks were constructed using the TCS algorithm in the population analysis with reticulate trees (POPART) (*Leigh & Bryant, 2015*). The POPART haplotype networks are illustrated in Figs. 5 and 6. Group A comprised eight haplotypes, with 307_Himachal Pradesh and 303_Haryana as the most distant haplotypes. Group B had seven haplotypes; the most distant haplotype was from Tripura. Some states had more haplotypes, such as Maharashtra (KF724977.1_Maharashtra), Delhi (MN594551.1_Delhi), and West Bengal; this may be an indication of population stabilization or successful niche formation in the species of these regions. Further research with larger datasets is necessary to examine the diversity and molecular evolution in these geographical regions.

## DISCUSSION

*T. tabaci*, commonly known as onion thrips, represents a complex of cryptic species with diverse genetic backgrounds. These cryptic species exhibit subtle morphological differences, making it challenging to use traditional identification methods. Characterizing crypticness within *T. tabaci* is crucial because of the potential implications for agriculture and pest management. The existence of different haplotypes and genotypes within a species can significantly influence various aspects of its biology, including host preference, reproductive capacity, resistance to pesticides, and vector competence for plant viruses (*Gawande et al., 2017*; *Jacobson & Kennedy, 2013*; *Li et al., 2014*; *Toda & Morishita, 2009*). Understanding the genetic diversity and population structure of cryptic species within *T. tabaci* can aid the development of more effective control strategies tailored to specific
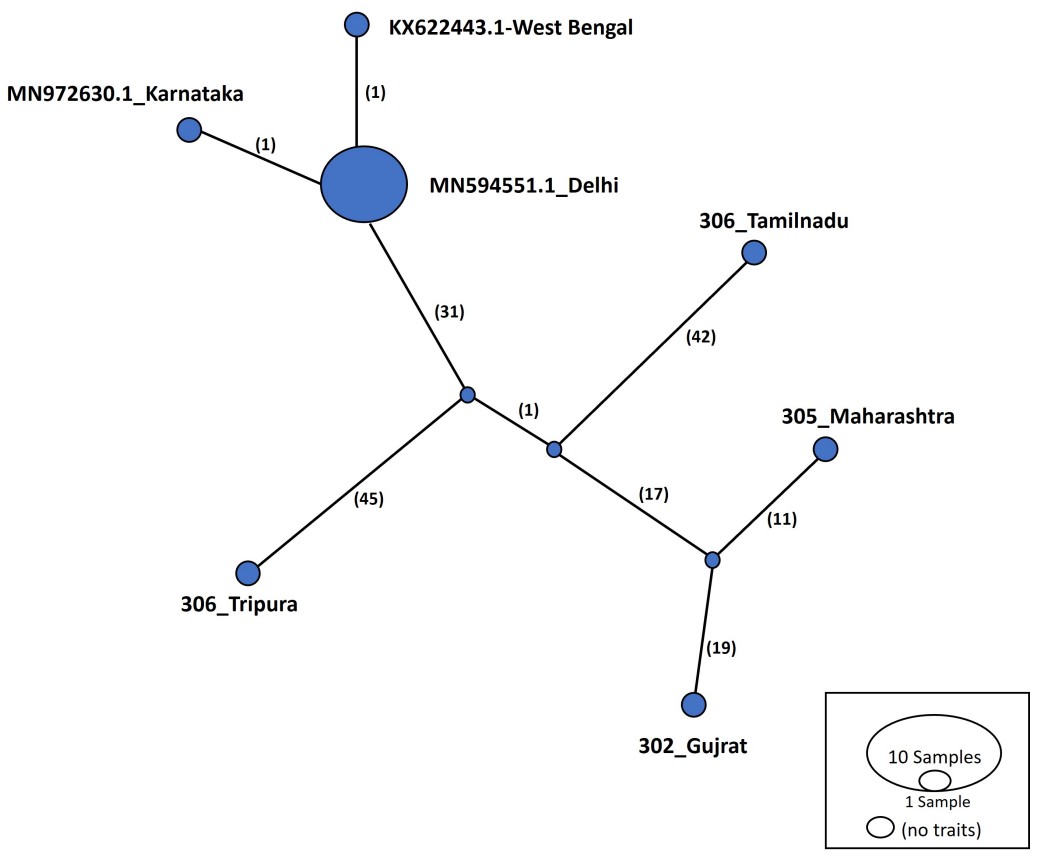

**Figure 6** **Haplotype network visualized in POPART for group B.** Numbers in bracket on the edges signify number of mutations and MN594551.1-Delhi represents predominant haplotype. Network has 7 haplotypes.

genetic variants. Furthermore, accurate identification of cryptic species is essential for monitoring population dynamics, tracking invasive species, and implementing quarantine measures to prevent the spread of economically damaging thrips biotypes. In this study, we attempted to characterize the crypticness within Indian *T. tabaci* population based on mtCOI gene sequencing for elucidating its evolutionary history. We determined that significant differences across the selected localities may be attributed to a combination of factors, namely geographical isolation, migration and gene flow, ecological variation, historical factors, and host plant associations.

The phylogenetic tree generated in our research showed a high degree of diversity, with new and a large number of gene sequences, suggesting that Indian thrips isolates continue to evolve and diversify. Higher intraspecific genetic variations in mtCOI gene of *T. tabaci* have been reported in economically important vegetables and crops host from different countries (*Kadirvel et al., 2013*; *Li et al., 2020*).

Nucleotide sequences of mtCOI gene were used for population genetics analysis of onion thrips from eight regions in India. This helped characterise species diversity, geographical distribution of a species, and tracing the phylogeographic origin (*Lombaert et al., 2014*;
*Porta et al., 2023*). In contemporary literature, only 28 haplotypes of mtCOI gene of *T. tabaci* have been reported worldwide, including Asia, Europe, Australia, America and other 23 countries, and these haplotypes were retrieved from nearly 282 nucleotide sequences (*Iftikhar et al., 2016*). This study investigated SNPs within the partial COI gene and differentiated *T. tabaci* populations into 30 mtDNA haplotypes, with 17 and 13 haplotypes in groups A and B, respectively. This indicated higher genetic variation or diversity of haplotypes in the Indian population of *T. tabaci.* The mtCOI region of thrips from Palampur represented the highest nucleotide polymorphism, followed by Tamil Nadu, Maharashtra, Haryana, and Gujarat. The high diversity in these locations may be attributed to climatic conditions and ecological factors such as local flora and fauna. Mitochondrial genetic variation observed in Delhi isolates was found to be lower than those in isolates from other regions. This can be attributed to several factors associated with the urban landscape of the region. Unlike other localities included in the study, Delhi's urban environment lacks the floral and fauna diversity that could influence the genetic makeup of the local *T. tabaci* population. A comprehensive analysis of SNP patterns within the mtCOI gene across diverse Indian locales revealed significant intrapopulation genetic heterogeneity and its potential repercussions on gene functionality. The SNP at the nucleotide position 63 (thymine to adenine transition) demonstrated remarkable allelic variation, indicating substantial genetic divergence among Indian cohorts. This phenomenon, in parallel with analogous allelic fluctuations at nucleotide positions 7 (cytosine to adenine transversion) and 399 (thymine to guanine transversion), accentuates the intricate mosaic of regional genetic variability. The notable allelic diversity at specific loci, particularly at position 63 in Rajgurunagar, Maharashtra, and position 7 in Palampur, Himachal, and Lembucherra, Tripura, suggest the possible adaptive evolutionary mechanisms in response to local environmental or nutritional selective pressures. These loci represent critical sites for probing the evolutionary trajectory and functional integrity of the COI gene. Distinct allelic distributions in regions such as Palampur, Himachal, and Coimbatore, Tamil Nadu, could mirror unique historical migration patterns or adaptive responses. Furthermore, the pronounced allelic variability at selected sites highlights the possibility of consequential impacts on COI gene transcriptional activity or protein efficacy, potentially affecting mitochondrial operational efficiency and predisposition to mitochondrial disorders. Nonetheless, loci exhibiting minimal allelic variation might reflect a shared ancestral genetic foundation, underscoring the dynamic balance between genetic diversification and evolutionary preservation.

Our study represents a comprehensive sequence data analysis with respect to the aspects of molecular phylogeny and genetic diversity. To achieve the computational results of genetic diversity, groups A and B were formed considering variations in the length of the mtCOI gene of onion thrips. High Nm values indicating high gene flow were observed more frequently across all populations in Group A than in Group B, indicating that these groups had recent population bottlenecks with varying Hd and Pi values (*Grant & Bowen, 1998*). Small Hd and Pi values (0.5 and 0.005, respectively) indicated a recent population bottleneck or founder event by a single or a few mtDNA lineages; high Hd (>0.5) and low Pi values (0.005) indicated a population bottleneck followed by rapid

population increase and mutation accumulation; low Hd (0.5) and high Pi (>0.005) values represented geographically subdivided populations; high Hd (0.5) and Pi values (>0.005) indicated a large stable population with a long evolutionary history or secondary contact between distinct lineages (*Grant & Bowen, 1998*). This helped us determine that genetic drift might lead to changes in diversity and the number of haplotypes (*Balloux, Lehmann & de Meeûs, 2003*). *T. tabaci* populations from the central and southern regions of India exhibited high levels of genetic differentiation and diversity compared with populations from other regions, whereas the population from the north-eastern region showed low genetic flow. Geographical isolation and absence of recent gene flow suggested that genetic drift played an important role in genetic differentiation. The findings of this study extend our knowledge on diversity of *T. tabaci* in India and provide insights for management practices to mitigate its growth.

Pairwise *Fst* and gene flow (Nm) among different zones, namely west, south, east, and north regions of Indian subcontinent, were calculated. Significant differences were observed between groups A and B, where Group A comprising populations from the southern and western parts of India and Group B, comprising populations from the southern parts, showed high Nm values. Nm < 1 reflected low levels of gene flow and ultimately higher genetic variation such as in north-eastern and northern parts (*Tiroesele et al., 2014*). Varying levels of gene flow may be attributed to the geographical and genetic distance. Fst values were <0.5 for majority of the samples, which is a relative measure of genetic variation that revealed overall similarity. For Group A, Fst was <0.5, suggesting a low level of genetic variability.

Haplotype networks generated in this study suggested that diverse haplotypes exist in Delhi and Maharashtra. These results are also consistent with those of molecular phylogeny, which indicate that geographical proximity cannot be used to measure molecular evolutionary patterns in mtCOI. Recent investigations in *T. tabaci* populations inhabiting on *Alliums* did not determine any correlations between the genetic and geographic distances (*Li et al., 2020*; *Porta et al., 2023*). The haplotype networks showed that Group A comprised eight haplotypes, with 307_Himachal Pradesh and 303_Haryana as the most distant haplotypes. In Group B with seven haplotypes, the most distant haplotype was from Tripura. Some states represented more haplotypes, like Maharashtra (KF724977.1_Maharashtra), Delhi (MN594551.1_Delhi), and West Bengal, which may be an indication of population stabilization or successful niche formation by the species.

Studies have reported links between traits, reproductive modes, and distribution of genetic variation from different countries, where, in particular, the mode of reproduction of thrips population affects their diversity (*Westmore et al., 2013*; *Gawande et al., 2017*; *Jacobson & Kennedy, 2013*; *Nault, Kain & Wang, 2014*; *Jacobson et al., 2016*). In this study, we could not highlight the presence of arrhenotokous and thelytokous populations, which could have helped rule out low or high genetic diversity to further explore Indian thrips.

## CONCLUSIONS

To the best of our knowledge, this study marks the first attempt to explore the genetic structure of *T. tabaci* populations. Amplicon sequencing of the mtCOI gene was used for

the analysis. Our investigation reaffirmed the previous findings that *T. tabaci* isolates in the Indian subcontinent exhibit significant diversity.

Our findings unveil multiple insertions and deletions occurring at various genomic positions across different localities, with the highest degree of variation observed within the 300–400 genome position range. Analysis of SNP patterns within the mtCOI gene across diverse Indian locales revealed significant intrapopulation genetic heterogeneity and its potential repercussions on gene functionality. Molecular diversity analyses identified a total of 30 haplotypes within the population, with noticeable gene flow occurring between specific subpopulations. Higher Fst values within northern-western subpopulations indicated considerable genetic variability, particularly evident in haplotype networks predominantly originating from the northern region, notably Delhi. While most populations exhibited stable and ancient evolutionary histories, the thrips populations in northern-western and north-eastern India seemed to indicate rapid growth.

### Funding
This study was supported by the India Council of Agricultural Research (ICAR), New Delhi: Project: Biotechnological approaches for biotic stress management (Project No. IXX16061). The funders had no role in study design, data collection and analysis, decision to publish, or preparation of the manuscript.

### Grant Disclosures
The following grant information was disclosed by the authors:
India Council of Agricultural Research (ICAR), New Delhi: IXX16061.

### Competing Interests
Suresh Gawande is an Academic Editor for PeerJ.

### Author Contributions
- Tushar Gawai performed the experiments, prepared figures and/or tables, and approved the final draft.
- Sharwari Sadawarte performed the experiments, analyzed the data, prepared figures and/or tables, and approved the final draft.
- Kiran Khandagale performed the experiments, analyzed the data, prepared figures and/or tables, authored or reviewed drafts of the article, and approved the final draft.
- Anusha Raj analyzed the data, authored or reviewed drafts of the article, and approved the final draft.
- Abhijeet Kulkarni conceived and designed the experiments, analyzed the data, authored or reviewed drafts of the article, and approved the final draft.
- Durgesh Kumar Jaiswal performed the experiments, authored or reviewed drafts of the article, and approved the final draft.
- Avinash B. Ade conceived and designed the experiments, authored or reviewed drafts of the article, and approved the final draft.

- Suresh Gawande conceived and designed the experiments, authored or reviewed drafts of the article, and approved the final draft.

## Data Availability

Raw data are available in the Supplemental Files.

Data is also available at NCBI SRA: PRJNA917098.

## Supplemental Information

Supplemental information for this article can be found online at http://dx.doi.org/10.7717/peerj.17679#supplemental-information.

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
