# Peer review of "Phylogenetic and population genetic analyses of Thrips tabaci Lindeman (Thysanoptera: Thripidae) on Allium host in India"

_PeerJ, doi:10.7717/peerj.17679_

## Round 0.1 · original submission · Major Revisions

All three reviewers have raised several critical issues and recommended substantial revision. Hence, authors are requested to resolve the reviewer's concern and submit a revised manuscript accordingly.

Reviewer 1 ·

Basic reporting

The research is very good, as it includes the genetic differentiation of a species of insect that is of economic importance, and it usually requires specialized taxonomists to identify it reliably.
But I two notes:
First: Classical taxonomy is extremely important and remains the basis for the identification process of species. Therefore, authors must include a section in which they talk about the most important diagnostic characteristics of the thrips species under study and mention the overlapping or difficult characteristics that require barcoding.
Second: This study represents the first of its type in India (and this is of course very good), but a comparison and discussion must be made with a recent study from a short period ago that deals with the same topic on a global level.
Porta B, Vosman B, Visser RGF, Galván GA, Scholten OE (2023) Genetic diversity of thrips populations on Allium species around the world. PLOS ONE 18(8): e0289984. https://doi.org/10.1371/journal.pone.0289984

Experimental design

Very good and enough

Validity of the findings

adequate within the boundaries of the study objectives

Additional comments

In addition to the notes in the basic report
There are other notes in an attached table

Thank you very much

Annotated reviews are not available for download in order to protect the identity of reviewers who chose to remain anonymous.

·

Basic reporting

The overall language of the manuscript is acceptable. However, it is crucial to enhance the literature citation and presentation, particularly by incorporating relevant findings related to Thrips tabaci and related species.

I recommend including references to important previous studies, such as the recent publication by Porta et al. (2023), titled "Genetic diversity of thrips populations on Allium species around the world" (DOI: 10.1371/journal.pone.0289984). Additionally, consider referencing the work of Bravo Pérez et al. (2018) on the species diversity of thrips in selected avocado orchards from Mexico, published in the Journal of Integrative Agriculture (DOI: 10.1016/S2095-3119(18)62044-1).

By integrating these relevant studies, you can provide a more comprehensive literature review and strengthen the foundation of your manuscript.

Experimental design

Methodology Clarity:
The methodology section, although citing previous methods, needs to provide more clarity and additional information to allow for the replication of the study by interested parties. Please consider providing detailed step-by-step procedures, specific conditions, and any modifications made to the cited methods. This will enhance the reproducibility of the study.

Abbreviation Clarification:
Abbreviations, such as SNPs, should be elaborated upon when first introduced in the manuscript. Even if they are widely recognized, providing a brief explanation upon initial use ensures clarity for readers who may not be familiar with the abbreviation.

DNA Extraction Elaboration:
The DNA extraction process requires more detailed explanations. Specify the number of aphids used for extracting DNA in one sample. Clarify whether a single adult, mixed culture, or multiple adults were used for DNA extraction. Providing this level of detail will contribute to the transparency of your methodology.

Please address these points to improve the overall clarity and completeness of the methodology section.

Validity of the findings

Comprehensive Study:
The study is commendable for its comprehensiveness, and novel attributes have been appropriately indicated. The presentation of sufficient data is appreciated.

Clarity in Presentation:
While the data is presented, there is room for improvement in providing additional information and clarity for each sub-study and analysis. Consider expanding on the details of each sub-study, providing more context, and ensuring a clear flow of information.

Discussion Enhancement:
The discussion section requires improvement by incorporating recent literature. Ensure that recent findings and relevant studies are integrated into the discussion to strengthen the overall argument and provide a more nuanced understanding of the results.

Conclusion
The conclusion is comprehensive; however, it would be beneficial to emphasize and highlight the major findings more explicitly. Consider providing a concise summary of the key outcomes, ensuring that the most significant results are clearly and prominently presented. This will assist readers in quickly grasping the primary contributions of the study.

Additional comments

While writing the manuscript, I noticed that the language flow is inconsistent. I would appreciate it if you could review and improve the flow for greater clarity.
In abstract
Background:
Clarify the term "cryptic species" for better understanding. Briefly explain how these species impact virus transmission.
Consider providing a concise definition or context for "transmission ranges for reproduction and hosts."
Methods:
Mention the rationale for selecting the eight diverse localities in India for amplicon sequencing.
Results:
Clearly outline the types and significance of Insertions and deletions observed in the genomic positions.
For molecular diversity analysis, briefly explain the relevance of identifying haplotypes and their distribution.
Discussion:
In the genetic variability discussion, elaborate on the potential reasons for the observed high levels of genetic variability in the northern-western subpopulation.

Consider providing a brief context on the importance of studying Thrips tabaci genetic diversity in the broader context of disease management.

Reviewer 3 ·

Basic reporting

Dear authors and editor,

While the manuscript holds value, there are significant issues, particularly in the analysis of phylogenetics and the presentation of tree/network illustrations. Improvement is essential.

Consider using alternative programs as MEGA proves to be unstable for analysis. The explanation of SNP analysis is lacking.

Figures are poorly integrated into the manuscript; kindly replace them for the second review. Utilize the R program for statistical analysis and graph/chart creation.

A Bayesian analysis for comparison would enhance the paper.

Note that all phylogenetic trees lack correctness and support values (e.g., bootstrap, aLRT, etc.). A major revision is necessary.

Regards,

Experimental design

Above stated

Validity of the findings

Above stated

Additional comments

Above stated

---

## Round 0.2 · Minor Revisions

Authors are requested to revise the manuscript as per suggestions of reviewer 3.

Reviewer 1 ·

Basic reporting

Dear Editor,

I see that the authors responded well to all the questions and recommendations in the previous review and that the artcile can be published in its current form.

Thank you very much

Experimental design

No comment

Validity of the findings

No comment

Additional comments

Thank you very much

·

Basic reporting

The manuscript has been significantly improved by addressing all the provided comments. The authors have successfully incorporated recent references, which enhance the relevance and credibility of the study. The revisions have clarified the methodology, strengthened the discussion, and ensured the conclusions are well-supported by the data. I appreciate the thorough efforts and contributions of the authors in advancing the research in this field.

Experimental design

The experimental design has been notably improved, providing greater clarity and detail.

Validity of the findings

The validity of the findings has been improved, along with the data presentation and clarity of the figures. I appreciate the authors' efforts to enhance the validity of the study and the overall quality of the manuscript.

Reviewer 3 ·

Basic reporting

Dear Editor and Authors,

I noticed that you have improved your manuscript, which sounds good! However, you still need to have the English language revised by a fluent speaker. Additionally, Figures 3 and 4 should not be represented as circular trees since your dataset is small. Instead, they should be changed to simple phylogram trees.

Once these changes are made, the manuscript could be accepted. Of course, you should also consider and incorporate the comments from other reviewers.

Best regards,

Experimental design

NA

Validity of the findings

NA

---

## Round 0.3 · accepted · Accept

The authors have incorporated the reviewer's suggestions. Hence, the revised manuscript is acceptable in its current state.